# OpenReview forum: "A Codespace Autoencoder for Language Tasks"
_ICLR.cc/2025/Conference — ICLR 2025 Conference Withdrawn Submission_

### Official Review · Reviewer_tJ91 · 2024-10-31

**Soundness:** 1
**Presentation:** 2
**Contribution:** 2
**Rating:** 3
**Confidence:** 3

**Summary:**

This paper proposes an LLM COT-based autoencoder for representing neurosymbolic programs. This autoencoder uses an LLM to translate a given natural language task description into a program that computes the output. The program is then executed (using a combination of actual execution and a neural network) to produce the output corresponding to the natural language task description. In this formulation, it is possible to sample correct input/output examples in a way that should have some correctness properties.

**Strengths:**

The main strength of this approach is that programs are far more compositional than text-based LLMs and thus this technique should be capable of scaling better than direct text-based approaches. In addition, LLMs are often capable of writing programs to solve tasks they would find difficult to do directly. Thus, in concept, this approach should improve correctness of generated data on tasks involving LLM-unfriendly tasks.

**Weaknesses:**

The main weakness of this paper is that it does not correctly summarize the largely negative results found in the Results section. The abstract and introduction should both mention that the Latent Code technique does not have the highest correctness or produce the best training data. Greater discussion is necessary in Sections 7.2 and 7.3 as to why this is the case. Additionally the conclusion does not appropriately summarize the results. “We show that code being inherently modular and executable offers advantages in correctness, particularly for algorithmic tasks” does not summarize the results section, which shows that the sampling baseline often overperformed Latent Code on these tasks.

I would like to emphasize that I do not think negative results alone are a reason to reject a paper; however I do believe that these results need to be appropriately contextualized.

Additionally, the equation on 85-89 is unclear to me, see Questions section for details.

Some other concerns:

I do not understand what is meant by Line 219. I assume the goal is not to have every z_i be the literal string “*So the answer is*”. I think it would be helpful to have a more concrete example or set of examples.

On line 241 you mention that a natural language input t_s is proposed for the sampled program. If this is via an LLM it would be useful to write this explicitly. Earlier, you set p(t | z) = 1, because it is known, so did not discuss how this probability might be computed or sampled from.

Minor nits:
62: “composed of modular function”; function -> functions.
77: y being an element of x is somewhat confusing. Perhaps either x or y could be renamed
209: call the section “Latent CoT” or update future references to Latent CoT to instead be “Textspace Autoencoder”
In 7.1 it might be best to add a parenthetical or a footnote to the effect of “what is being compared here is y, since we compute P(t | z) = 1”
Table 2: call the “Sampling” baseline “interpolation” to match the text.

**Questions:**

The mathematical derivation on 85-89 is unclear to me. The first step makes sense, but the second step seems to have two errors
1. The sum over i seems to disappear, however I believe it should still be there in the final equation
2. The denominator going from p(z_i | z_L, x_i) to p(x_i | z_i, z_L) p(z_i | z_L) seems incorrect. Doing the math out, p(z_i | z_L, x_i) = p(z_i, z_L, x_i) / p(z_L, x_i) = p(x_i | z_i, z_L) p(z_i, z_L) / p(z_L, x_i) = p(x_i | z_i, z_L) p(z_i | z_L) p(z_L) / p(z_L, x_i) =  p(x_i | z_i, z_L) p(z_i | z_L) / p(x_i | z_L). There is an extra p(x_i | z_L) term in there. If this is a constant it requires a bit more justification in the text.

I think it would be helpful to include a more detailed derivation that does not elide these steps, allowing a reader to more closely follow the derivation.

---

### Official Review · Reviewer_TaWt · 2024-11-03

**Soundness:** 3
**Presentation:** 1
**Contribution:** 3
**Rating:** 3
**Confidence:** 2

**Summary:**

This paper presents a method of learning an "autoencoder" for language tasks using in-context learning with large language models. Specifically, the latent space of the autoencoder consists of a (pseudo-)code representation of the language task, which when executed on a (natural language input), produces the desired output. The optimization of the autoencoder proceeds by searching for a discrete library of functions. The results show that, for both a code- and CoT-based latent space, the learned latent space improves the faithfulness of reconstruction and correctness when sampling from the autoencoder.

**Strengths:**

The approach strikes me as a very interesting and novel way to leverage LLMs to improve LLMs by learning domain-specific knowledge in context. The results are also suggest improvements over baselines on the evaluated metrics. I found Table 2 to be most interesting in that using a code-based "latent" representation was better for algorithmic tasks, while pure CoT is better on non-algorithmic tasks.

**Weaknesses:**

Unfortunately, I found the details to be extremely unclear, which ultimately makes it difficult for me to provide an accurate assessment of the contributions (and advocate for acceptance).

## Clarity

A number of different symbols are introduced without a definition. As a result, I could not grasp many of the critical pieces completely: sampling from the posterior, what is being learned during optimization, and how the metrics are defined and evaluated.

In my opinion, the presentation could be substantially improved by providing examples alongside the notation. For instance, give an example of $x$, $t$, $y$, illustrate one sampling from $q$, etc.

## Significance

As far as I can tell, applications of the approach are limited to sampling synthetic data which are more domain-appropriate and correct. However, the data could be correct simply because the generated task is much easier. No experiments are run that train on this synthetic data, hence we do not know if the synthetic data actually improves anything for downstream models.

It could be interesting to explore if the method could be used to improve correctness on a given dataset (i.e., improve reasoning capability of LLMs via library learning and a form of automatic prompt optimization).

**Questions:**

- line 96: what is $z$, $\bar z$ and $\tilde z$?
- line 192: why is the text input given in $P(t | z)$? I thought $z$ was the "latent" (or the code representation)
- line 197: what is z(t)?
- How is recovery rate in Table 1 defined (as a mathematical equation)
- What is the library initialized to?

---

### Official Review · Reviewer_w7cW · 2024-11-03

**Soundness:** 3
**Presentation:** 2
**Contribution:** 2
**Rating:** 5
**Confidence:** 3

**Summary:**

The paper presents a novel autoencoder framework, the Codespace Autoencoder, which extracts latent code representations from textual data. The method is non-parametric, utilizing in-context learning and code interpretation for inference. The paper demonstrates the effectiveness of this approach through experiments on the Super-NaturalInstructions dataset, comparing it with other latent modeling approaches and showing its superiority in generating correct and domain-relevant text, especially for algorithmic tasks.

**Strengths:**

1. The use of code as a latent representation is a novel idea.
2. The induced latent space is structured, allowing for modular exploration and compositional decoding into text.

**Weaknesses:**

1. While the paper presents promising results for algorithmic tasks, the proposed approach may be less effective for less structured tasks that lack a clear symbolic representation, which are more prevalent in practice.
2. The process of inducing the latent space and generating synthetic data renders this method more computationally intensive compared to traditional approaches.
3. In the downstream training results, data generated from the sampling baseline outperforms data sampled from the latent modeling method. This raises a significant concern: how can the method be considered valuable if its downstream training results are inferior?

**Questions:**

See weaknesses.

---

### Official Review · Reviewer_QEVo · 2024-11-03

**Soundness:** 1
**Presentation:** 2
**Contribution:** 2
**Rating:** 3
**Confidence:** 3

**Summary:**

- This paper proposes a method for extracting latent symbolic representations for text-generation datasets based on their programmatic structure. The work is motivated by the observation that many tasks specified in natural language encode underlying program-like structure.
- The authors describe a method for mapping datasets to task-level libraries of functions that can be used to generate new tasks via posterior sampling. This approach is framed as a special case of variational autoencoding where the latent space consists of discrete programs, both the encoder and decoder are non-parametric (i.e., frozen pre-trained LLMs), and inference is performed using in-context learning.
- Experiments are performed on a subset of the Super-NaturalInstructions dataset (SNI; Wang et al., 2022). The proposed method is compared against two baselines: an “interpolation” baseline that uses LLM prompting with no latent representation, and a chain-of-though (CoT) baseline that uses natural language reasoning traces instead of code as the latent representation. Comparisons are performed using two LLMs (Llama3.1 8B and Mixtral 8x22b) in two experiment settings. In the first experiment, which focuses on reconstruction accuracy, tasks are evaluated for correctness %, domain relevance %, and diversity (using cosine similarity). In the second experiment, synthetic tasks sampled from the different methods are used to finetune a Pythia 1.4B model, which is evaluated on OOD examples. Finally, some qualitative analysis is presented on examples of programs, execution traces, and library functions generated by the proposed method.

**Strengths:**

- The proposed approach is well-motivated by the intuition that many tasks specified in natural language encode underlying program-like structure.
- On a purely theoretical level, the framing of the approach is novel. Specifically, while there exists prior work learning libraries of functions for natural language and reasoning tasks, to my knowledge, this is the first work to frame this process using the formalism of variational autoencoding using LLMs.
- Since many natural language tasks cannot actually be reduced to symbolic programs, the authors propose a hybrid execution model where lines of code that fair to execute with a Python interpreter are emulated via a LLM query (as in Li et al., 2024). This allows the latent representation to include “imaginary” functions whose concrete semantics are undefined. This represents an interesting and potentially powerful new execution model, the properties of which are relatively unexplored.
- In reporting the results, the authors are relatively transparent about instances where the proposed method does not perform favorably compared to baselines.

**Weaknesses:**

**Statistical soundness:** Various comparisons are made between methods based solely on the mean statistic (e.g., Tables 1-3). Without significance testing, all comparisons between the proposed method and baselines are without statistical basis. This is especially an especially pressing issue due to the small sample sizes in these evaluations—for instance, Table 2 is based on just 3 tasks. **I strongly recommend that this paper not be accepted for publication until the authors perform significance testing and report 95% confidence intervals to support all between-methods comparisons.**

**Lack of compelling results:** Even looking at only mean statistics, it is difficult to say that the Latent Code approach actually helps over baselines. It seems that there is a tradeoff—as the authors note, “using code produces more correct data in algorithmic tasks and using CoT produces more correct data in non-algorithmic tasks” (L363-364). In downstream training experiments, the proposed method apparently does not help at all; the authors observe that “data from the sampling baseline generally outperforms data sampled from the latent modeling method” (L373-374). While I commend the authors for plainly reporting mixed results and failure cases, in the current arc of this paper, the Result section (Sec. 7) comes as a big letdown. In order to compel readers to invest the time to understand the details of the proposed method, there needs to be a clearer and more favorable results story.

**Narrowness of evaluation domain:** All evaluations in this work were performed on a single dataset, Super-NaturalInstructions (SNI) (Wang et al., 2022). While the authors characterize this dataset as “diverse,” they also only consider a subset covering only approx. 13% of the tasks (200 / 1616) in SNI. It would be nice to see evaluations on a broader subset—better still would be to demonstrate that the proposed method generalizes to other datasets beyond SNI.

**Human evaluations:** Human evaluation metrics are reported for correctness and domain relevance (Table 2) but I was not able to find any details about how these human evaluations were conducted in the main paper or in the appendices. Were these performed by recruiting subjects on an online evaluation platform or were these conducted by the authors? What instructions were provided to the evaluators? What were the rating scales? What steps were taken to ensure that the evaluations were unbiased (esp. if the authors themselves performed these evaluations)? At minimum these details should be included in an appendix. I also noted that human evaluations were only performed for 1/3 methods in Table 2. For completeness and fairness of comparison, I suggest that these evaluations be performed for all methods in Table 2.

**Qualitative analysis is not systematic:** Sec. 8 shows a handful of examples of programs, execution traces, and top functions. However, this analysis feels haphazard and rushed: it’s unclear how these specific program examples were selected, and in some cases (e.g., Listing 3) it’s also unclear what they are meant to demonstrate. The final part of Sec. 8 (Does the shared structure of the latent space reflect underlying patterns of the task?) is intended to show how the frequently used functions correspond to the task structure, but only the top-2 functions are shown for only a small handful of tasks. Moreover, while some of these are standard Python functions (e.g., `re.sub`), many are task-specific functions whose semantics are completely opaque to the reader. (i.e., Is the `complete_story` function imaginary or real? Either way, just because the LLM named a function `complete_story` in response to the task “Write the middle sentence of a story” doesn’t mean that the function is actually *useful* for solving the task, or that an LLM asked to emulate this function will do so reliably while respecting other constraints.) More thought and attention is needed to turning this section into something that conveys to readers a clear picture of what the latent space / learned libraries actually look like.

**Superficiality of autoencoder formalism:** The proposed method is presented as a “non-parametric latent model” dressed in the traditional formalism of variational autoencoders (Sec. 2). However, because the latent space here is discrete, traditional methods for optimizing VAEs with continuous latents are not applicable, necessitating combinatorial optimization. As the authors note (L102), this search problem is intractable, so instead, they use rejection sampling over in-context examples. This is a reasonable choice, but there exist more principled and efficient approaches to combinatorial search over programs, such as those used in prior work based on e-graph matching (Ellis et al., 2021; Cao et al., 2023) or branch-and-bound search over program ASTs (Bowers et al., 2023). Moreover, it means that much of the autoencoder formalization is just window dressing; and consequently, the present work is a lot more similar to discrete approaches to library learning than it is to prior work on autoencoders.

**Missing citations:** The related works (Sec. 3) is reasonably inclusive, but is missing citations to two highly relevant papers:

- **LILO: Learning Interpretable Libraries by Compressing and Documenting Code (Grand et al., 2024).** This paper was presented at the most recent ICLR conference. LILO also learns libraries of reusable functions that can be used to solve tasks by writing instance-level programs. Both works use similar in-context learning methods with LLMs and share similar evaluation settings. The key conceptual difference is in the framing of the problem: LILO follows in a line of work that formalizes library learning as symbolic compression (Ellis et al., 2021; Wong et al., 2021; Bowers et al., 2023). In contrast, the present work instead reframes this process using autoencoders (though as discussed above, this framing is somewhat superficial as it reduces to combinatorial search in practice). In any case, given the clear conceptual and methodological resemblances, LILO is a key related work on learning latent symbolic descriptions of datasets using LLMs that should be cited and discussed appropriately.
- **Representing Partial Programs with Blended Abstract Semantics (Nye et al., 2021).** This work also describes an approximate program execution model using neural networks (referred to in this work as “emulation”). Whereas Nye et al. use learned neural modules, the present work uses pre-trained LLMs with in-context learning.

## Other nits

- Fig. 1: It would be helpful to label each of the boxes; e.g., “Input task description”, “Library”, “Instance-level program”
- It would be nice for the abstract/intro to preview the evaluations and results a bit more clearly: what datasets / tasks are part of the evaluations, and what are the key results?
- “we are interested in finding a latent symbolic representation $z_i \in \mathcal{Z}$” (L79-80): $\mathcal{Z}$ is not properly defined in the text and it was not immediately clear at this early point in Sec. 2 whether the $z_i$ terms were symbolic functions, real-valued vectors, or something else.
- Table 2: GPT-4o is listed, but these evaluations were actually performed with GPT-4o-mini. I’m assuming that the model name was abbreviated for space, but this seems like an unreasonable presentational choice as there are non-trivial differences between the capabilities of these two models as evaluators. I recommend that the column headings be renamed to GPT-4o-mini — it does seem like there is enough horizontal whitespace available.
- “Diversity is measured with average embedding cosine similarity to centroid per task” (L346): Doesn’t this mean that higher is better for the avg. cossim statistic? Also, this statistic should be clearly labeled as corresponding to “Diversity” in Table 2.
- In Table 3, results from two different LLMs (Mixtral and Llama) are intermixed in a way that makes comparing rows of the table unnecessarily difficult. Could this table be reformatted to group by LLM type (as in Table 2)?
- In the text, the internet query function is called `internet_lookup()`, but in the code examples in Sec. 8 it is called `lookup_on_internet()`

**Questions:**

1. “We approximate reconstruction accuracy $p(y | z, t)$ with text similarity metric threshold $γ$” (L195; see also Lines 284-287.): How was this threshold determined?
2. What was the motivation for introducing an `internet_lookup()` function? Which tasks in SNI require internet lookup beyond the internal knowledge of the LLMs? It would be nice to see some analysis in Sec. 8 breaking down LLM emulation rates by frequency of internet query usage.

---

### Official Review · Reviewer_n88u · 2024-11-04

**Soundness:** 3
**Presentation:** 3
**Contribution:** 3
**Rating:** 6
**Confidence:** 4

**Summary:**

This paper introduces a novel autoencoder framework that models text data as being generated from underlying code with a dataset-level function library. The authors propose using code as a latent representation for text data, arguing that many natural language tasks have an underlying symbolic process that can be well-represented by code. The method is non-parametric and leverages in-context learning and code interpretation for inference. The authors evaluate their approach on the Super-NaturalInstructions dataset, comparing against baselines including direct sampling and using chain-of-thought as a latent representation.

**Strengths:**

- The paper is well-written and presents an interesting experimentation on modeling text data using code as a latent representation.
- The approach is well-motivated, with clear arguments for why code makes a good latent representation (modularity, executability)
- The analysis of LLM emulation rates provides useful insights into how the method works in practice.

**Weaknesses:**

- The method underperforms simple sampling baselines for downstream model training, as shown in Table 3.
- The approach seems to work better for algorithmic tasks than general language tasks (Table 2, Section 7.2), limiting its broad applicability
- The experiments only use two models (llama 3.1 8b and mixtral 8x22b), and make generalized claims such as "smaller models" when speaking about llama 3.1 8b results and "larger models" when speaking about mixtral 8x22b.

**Questions:**

- How might performance scale as the complexity (e.g. compositionality) increases?

---

### Note · Authors · 2024-11-26

I have read and agree with the venue's withdrawal policy on behalf of myself and my co-authors.